# Hypertriglyceridemia-Induced Acute Pancreatitis—The Milky Way Constellation—The Seven-Year Experience of a Large Tertiary Centre

**DOI:** 10.3390/diagnostics14111105

**Published:** 2024-05-26

**Authors:** Andrei Vicențiu Edu, Mihai Radu Pahomeanu, Andreea Irina Ghiță, Dalia Ioana Constantinescu, Daniela Gabriela Grigore, Andreea Daniela Bota, Daniela Maria Luta-Dumitrașcu, Cristian George Țieranu, Lucian Negreanu

**Affiliations:** 1Faculty of Medicine, Carol Davila University of Medicine and Pharmacy Bucharest, 050474 Bucharest, Romaniadalia.constantinescu@stud.umfcd.ro (D.I.C.); daniela.grigore@stud.umfcd.ro (D.G.G.); andreeadbota@gmail.com (A.D.B.); lucian.negreanu@umfcd.ro (L.N.); 2Internal Medicine and Gastroenterology Department, University Emergency Hospital of Bucharest, 050098 Bucharest, Romania; 3Bucharest Acute Pancreatitis Index (BUC-API) Study Group, 077135 Mogoșoaia, Romania; 4Gastroenterology Department, University Emergency Hospital Elias, 011461 Bucharest, Romania; marialutadumitrascu@gmail.com

**Keywords:** acute pancreatitis, hypertriglyceridemia, outcome, recurrence, ICU, severity

## Abstract

(1) Background: Hypertriglyceridemia (HTG) is a well-known metabolic condition associated with an increased risk of acute pancreatitis. In this study, we tried to establish whether there are any significant disparities concerning recurrence rate, intensive care unit (ICU) admission, hospital (ICU and total) length of stay (LoS), morphology, severity and age between HTG-induced acute pancreatitis and any other known cause of pancreatitis (OAP). (2) Methods: The research was a retrospective unicentric cohort study, using information from the Bucharest Acute Pancreatitis Index (BUC-API) registry, a database of 1855 consecutive cases of acute pancreatitis. (3) Results: We found a weak association between HTG-AP and recurrence. The HTG-AP patients were younger, with a median of 44.5 years, and had a longer ICU stay than the OAP patients. In addition, we identified that the HTG-AP patients were more likely to develop acute peripancreatic fluid collection (APFC), to be admitted in ICU, to have a more severe course of disease and to be cared for in a gastroenterology ward. (4) Conclusions: Hypertriglyceridemia-induced APs have a more severe course. The typical patient with HTG-AP is a middle-aged male, with previous episodes of AP, admitted in the gastroenterology ward, with a longer ICU stay and longer length of hospitalization, more likely to evolve in a severe acute pancreatitis (SAP) and with a higher probability of developing APFC.

## 1. Introduction

In the darkest hours of our nights, with the clearest skies, since the dawn of humankind, one can observe a milky river of stars above one’s head. No surprise that the Romans first called this the Via Lactea or the Milky Way [1]. What they did not know at that time was that what they saw was just a part of a very large galaxy of which we are also part. Eerily, in the same way, since the dawn of medicine, some physicians have observed that some patients’ serum has a milky appearance (see Figure 1). Later, with the advance of biochemistry, we were able to conclude that this macroscopic alteration of serum is due to hypertriglyceridemia [2].

Although sustained advances in the field of lipidology and pancreatology have been made, there are still several large gaps in the research regarding the HTG-AP constellation of demographical risk factors and clinical outcomes, again eerily similar to how astronomers are deciphering the mysteries of the Milky Way constellations which are visible to all but compressible only to some.

### 1.1. Acute Pancreatitis

Acute pancreatitis has an incidence roughly estimated [3] from 2.7 person-year/100,000 in Australia [4] to 134.9 person-year/100,000 in the USA [5] and 29.2 hospitalization-year/100,000 in Romania [6]. Acute pancreatitis is a disease that had a prevalence of 2.8 million cases worldwide annually as estimated by Global Burden of Disease Study 2019 [7].

According to the PREDATORR study [8], 67.1% of Romanian adults had at least one lipid abnormality. In particular regard to hypertriglyceridemia, the Romanian prevalence was estimated at 4.1%.

According to a review from 2019 [9], at least 20.2% of patients with hypertriglyceridemia had at least an episode of AP. The same review estimated a prevalence of HTG-AP at around 4% of total cases of AP.

The main etiologies are alcohol consumption, gallstones and hypertriglyceridemia [6]. Due to the lack of an efficient clinical risk assessment tool [10], AP has become an important financial burden for gastroenterological and surgical facilities around the word, with a study [11] from the USA estimating the cost to be US$2.2 billion in 2007 alone. Even though mortality in general AP cases is gradually decreasing [12], HTG-AP is associated with a more severe course of disease [9] and is probably triglyceride level dependent [13].

### 1.2. Hypertriglyceridemia

Hypertriglyceridemia is a common metabolic condition, with an estimated prevalence of 26% in the US adult population, or 12.3 million people, affecting predominantly white males [14]. The principal risk factors are obesity, sedentarism and diabetes [15].

Hypertriglyceridemia is a well-known cause of AP, with a systematic review by Carr et al. [16] conducted in 2016 concluding that 14% of patients with HTG will develop at least one episode of AP in their lifetime. In terms of classification, there are several triglyceride (TG) thresholds, as some sources consider severe any level >1000 mg/dL [9,17], while others consider severe any level >500 mg/dL [18,19].

Aside from dietary and physical activity measures, the treatment of HTG includes several pharmacological classes of fibric acid derivatives, omega-3 fatty acids and statins.

### 1.3. Study Aims

The primary aim of this study was to observe if there are any meaningful disparities in the number of recurrent episodes of pancreatitis when HTG-AP is compared with OAP. The secondary aims were to observe the differences in age, gender, severity, ICU admission rate and LoS-ICU, morphology as described by RAC [20], total LoS, type of ward of care, daily cost of hospitalization, outcome at discharge, rurality and tobacco smoking between HTG-AP and OAP.

## 2. Materials and Methods

### 2.1. Bucharest Acute Pancreatitis Index (BUC-API) Registry

This is a retrospective, large cohort study that employs data from the BUC-API registry, a single-center repository comprising 2039 cases of various pancreatic conditions, including acute pancreatitis (AP), recurrent AP and acute-on-chronic pancreatitis.

The creation of the BUC-API registry was approved by the Ethics Committee of the University Emergency Hospital of Bucharest, and all patients provided informed consent. The study was conducted in compliance with the ethical guidelines outlined in the 1975 Declaration of Helsinki and adhered to the STROBE guidelines.

The cases in the registry are sourced from 35 out of 42 counties in Romania, primarily concentrated (87.3%) in southern Romania, encompassing Bucharest and nearby counties.

Each new admission of a patient was treated as a separate case in this study. The cases were identified from the electronic health records (EHRs) of the University Emergency Hospital of Bucharest using specific ICD-10 codes (K85, B26.3 and B25.2) for consecutive discharges occurring between 1 June 2015 and 1 April 2022.

Initially, a total of 2520 cases were extracted from the EHRs. However, after screening by trained medical staff, 426 duplicate cases arising from software processing errors were identified. We considered 55 cases miscoded as they did not achieve at least two out of three criteria for diagnosing AP (three times the normal range of amylasemia or lypasemia, specific abdominal pain and image criteria). The miscoded cases were removed, leaving 2039 consecutive cases in the BUC-API registry. Of these, 184 (9.0%) were diagnosed with acute-on-chronic pancreatitis, but they were not identified as miscoded due to the absence of specific ICD-10 codes for this condition. The final count of consecutive AP cases in the registry was 1855 (91.0%). This registry represents the largest collection of pancreas-related disease data in Romania to date (see Figure 1).

The University Emergency Hospital of Bucharest (Romanian: Spitalul Universitar de Urgență București) is one of the largest acute-care tertiary teaching hospitals in Romania, with 1099 beds and housing both gastroenterological and abdominal surgery departments.

### 2.2. Case Selection

In the 1855 cases with AP obtained from the BUC-API registry, there were missing data for tobacco smoking (*n* = 1406, 75.8%), morphology (*n* = 562, 30.3%) and rurality (*n* = 16, 0.9%). Table 1 provides additional information about the characteristics of the patients, including the classification of the morphology and severity according to the Revised Atlanta Classification [17].

To simplify the statistical analysis, we focused only on the six most commonly encountered causes of AP, as outlined in Table 1. Whenever more than one etiology was reported, we reported the case as a single cause AP based on author consensus.

To define HTG-AP, we used the following hybrid criteria: serum triglyceridemic levels above 750 mg/dL at any moment of the hospitalization and no other obvious cause of AP. We mentioned hybrid criteria as the threshold for severe hypertriglyceridemia which is 500 mg/dL and requires urgent dietary and pharmacological intervention, and is mainly used by diabetologists [19] and cardiologists [18], but on the other hand, surgeons and gastroenterologists [9,17] usually follow a more liberal (>1000 mg/dL) threshold concerning the risk of developing AP secondary to HTG. We found this to be highly confusing, so we chose to establish a threshold based on their average. Cases of idiopathic AP were excluded from the comparative analysis.

Any case of AP without signs of chronic pancreatitis which had a prior hospitalization at our facility within the timeframe of the BUC-API registry or had explicitly mentioned previous episodes of AP in the EHRs, was considered a recurrence. All cost related data was initially calculated in RON and afterwards converted in EUR at the National Bank of Romania exchange rate rounded at two decimals from 16 May 2024 (1 EUR = 4.98 RON).

In order to measure serum triglycerides (TG), patients were fasted for 8 to 12 h prior to blood draw, venous blood samples were drawn (at least 0.5 mL), blood was transported at room temperature to the hospital laboratory via a Serum Separating Tube that contains silica particles and a serum separating gel. If there were more than 2 h between blood drawing and arrival to the laboratory, the blood sample was dismissed.

### 2.3. Statistical Tests

For this study, data organization was conducted using Microsoft Office Excel 2019©, now known as Microsoft 365© Version 2309 (Microsoft Inc., Seattle (WA), USA) and Google Docs© (Alphabet Inc., Mountain View (CA), USA). To examine the general characteristics of the cohort presented in Table 1, frequency tests were employed. Additionally, to explore the correlation between two categorical variables, Pearson’s chi-squared and Cramér’s V tests were performed. The Mann–Whitney U test was used to assess the correlation between a continuous and another categorical variable. All statistical analyses were performed using IBM SPSS Statistics Version 29.0.0.0© (IBM Inc., Armonk, NY, USA).

Results of the analysis with a *p*-value of <0.05 were considered statistically significant, while those with a *p*-value of <0.10 but greater than 0.05 were considered to have weak statistical significance. *p*-values were reported up to the third decimal only when they were close to the mentioned significance levels.

For the management of references in this paper, Zotero 6 for Windows and Zotero Connector Version 5.0.68 (Corporation for Digital Scholarship, Vienna, VA, USA) for Google Chrome Version 93.0.4577 (Alphabet Inc., Mountain View, CA, USA) were used.

## 3. Results

### 3.1. Population Characteristics

Most of the cases in our study were reported as OAP (*n* = 1514, 81.7%) with biliary (*n* = 732, 39.5%) and alcoholic (*n* = 628, 33.9%) prevailing, and HTG-AP accounting for 3.1% (*n* = 58). Most cases experienced a mild course of disease (*n* = 954, 51.4%).

The median age was 57.0 years (IQR = 26.0) with a majority of males (*n* = 1098, 59.2%) and a median LoS of seven days (IQR = 6.0). The median daily cost of hospitalization (DCH) was reported as 184.9 EUR (interquartile range (IQR) = 86.8).

We found several non-chronic recurrences (*n* = 319, 17.2%) in the study population, with an almost 1:1 ratio in relation to the type of ward of care.

Pertaining to the HTG-AP population, we found several cases of non-chronic recurrences (*n* = 18, 31.0%), a majority of males (*n* = 37, 63.8%) and cases mostly treated in gastroenterological wards (*n* = 42, 72.4%), but a younger median age (Md = 44.5, IQR = 9), a longer median LoS (Md = 8.3, IQR = 9.3) and a lower median DCH (Md = 167.5, IQR = 101.4).

We report an overall mortality of 5.8% (*n* = 108) with an HTG-AP mortality of 5.2% (*n* = 3).

We found the rate of ICU admission in the general population to be 9.6% (*n* = 179), with a higher rate amongst HTG-AP cases (*n* = 10, 17.2%). The median LoS in ICU in the general population was 3.0 days (IQR = 5.0), with a longer median LoS in ICU for HTG cases (Md = 7.0, IQR = 22.5).

Regarding their environmental risk factors, we found that most of the cases in the general population came from an urban environment (*n* = 1332, 71.8%), and this was also valid for HTG cases (*n* = 41, 70.7%). Although we took into account tobacco usage, there was a lack of data on this factor in the majority of cases, both in the general population (*n* = 1406, 75.8%) and for HTG cases (*n* = 41, 70.7%). Extensive details are provided in Table 1 and Table 2.

### 3.2. Single Attack or Recurrence?

For the primary aim, we ran the chi-squared test that showed a statistically significant association between etiology and recurrence (X2(1) = 6.9, *p* < 0.01). To determine the strength of the association, we used Cramér’s V test where +0.07 implied a weak association. Running the post-hoc test, we discovered an adjusted standardized residual (ASR) of +2.6 regarding the recurrence of HTG-AP, proving an important difference from the expected frequencies (see Table 2).

### 3.3. Severity

To determine the association between severity and etiology, we performed a chi-squared test on a three-level stratification of severity (as described by the Atlanta Revised Classification) that indicated a meaningful association between the two variables (X2(2) = 9.5, *p* < 0.01) with a weak association as observed by a Cramér’s V score of 0.08. Using post-hoc tests, we obtained a significant deviation from normal distribution, with an ASR of −3.0 regarding mild severity of HTG-AP. To refine our results, we combined the moderate severity and severe groups to determine whether HTG-AP is associated more with these two conditions. We again observed a meaningful association (X2(1) = 8.8, *p* < 0.01) with a weak association (Cramér’s V = 0.07), and the post-hoc test revealed an ASR of +3.0. This represents an important deviation from the expected frequencies that implies a tendency for HTP-AP cases to be associated with a more severe course of disease than OAP cases (see Table 2).

### 3.4. Age

To assess the relation between age and etiology, we ran a Mann–Whitney U test that revealed meaningful disparities (U = 22,753.5, Z = −6.2, *p* < 0.01). We are therefore able to report that HTG-AP patients (Md = 44.5, IQR = 9.0) are significantly younger than OAP patients (Md = 57.0, IQR = 25.0).

### 3.5. ICU

Related to ICU admissions, the chi-squared analysis revealed a meaningful association with etiology (X2(1) = 5.7, *p* = 0.02), with a weak association as revealed by a Cramér’s V score of 0.06. Post-hoc analysis revealed an important deviation from the expected frequencies, with an ASR of +2.4 in the case of ICU admissions of HTG-AP cases. The Mann–Whitney U test showed a meaningful difference in respect of LoS-ICU (U = 297.0, Z = −2.6, *p* < 0.01) as HTG-AP patients (Md = 7.0 days, IQR = 22.5) had longer ICU stays than OAP patients (Md = 3.0 days, IQR = 4.0) (see Table 2).

### 3.6. Length of Stay

We ran the Mann–Whitney U test to determine the relation between LoS and etiology. The test revealed a significant statistical difference between the two etiological groups (U = 35,278.0, Z = −2.6, *p* = 0.01), with longer stays for HTG-AP patients (Md = 8.0 days, IQR = 9.25) than for OAP patients (Md = 7.0, IQR = 5.0).

### 3.7. Morphology

Using chi-square analysis, we looked for any possible association between morphology and etiology and found a statistically significant association (X2(6) = 14.9, *p* = 0.02). For post-hoc testing, we saw an ASR of +2.3 about APFC and an ASR of −2.9 related to a normal pancreas, both results showing meaningful deviations from the expected frequency in the HTG-AP group (see Table 2).

### 3.8. Ward of Care

By performing chi-square analysis, we were able to find a significant association between etiology and ward of care (X2(1) = 12.1, *p* < 0.01), with a weak association (Cramér’s V = 0.09). Post-hoc testing revealed an ASR of +3.5 for HTG-AP patients in gastroenterological wards (see Table 2).

### 3.9. Other Aims

We ran chi-squared tests but did not find any significant associations between etiology and the following variables: gender (X2(1) = 0.2, *p* = 0.62), outcome at discharge (X2(4) = 2.6, *p* = 0.63), tobacco smoking (X2(2) = 4.8, *p* = 0.09) or rurality (X2(2) = 0.6, *p* = 0.74). The Mann–Whitney U test was performed to determine the association between etiology and DCH; however, no association was found (U = 29989.0, Z = −0.5, *p* = 0.59). See Table 2.

## 4. Discussion

The process by which HTG induces AP is not completely understood. The main hypothesis is that the large diameter of TG-rich lipoproteins, mainly chylomicrons, may disrupt the normal blood flow of the pancreas, resulting in ischemia which may trigger enzymatic activation within the acinar cells that promote further damage and generate proinflammatory free fatty acids. This uncontrolled inflammatory process may cause pancreatic edema, necrosis, and systemic inflammatory response syndrome (SIRS) [21].

In comparison with OAP cases, HTG-AP cases are prone to recurrence, with our findings being in congruence with several other retrospective and prospective studies on this topic [22,23,24,25]. We did find a rate of recurrence of 31.0%, which is similar to those of other studies on this topic as presented in a meta-analysis in 2023 [26]. This high rate of recurrence in HTG-AP might probably be related to the poor adherence of patients to the drug therapy prescribed. Better counseling and drugs with a longer half-time might alleviate this problem.

Our results indicate that a mild course of disease is associated with OAP. This implies indirectly that HTG-AP might be associated with a more severe course of disease. The results are in line with the results of a 2016 meta-analysis [16] but contradict the data from another retrospective study from the same year [27] as well as a narrative review published in 2018 [28].

Taking into account that Jo et al. [29] found that the severity of HTG-AP correlates with higher serum levels of TG, it will be interesting to see if whether one maintains a threshold of >1000 mg/dL [27,29] and then the association with severity is clear, or if one takes into consideration lower thresholds than ours (the aforementioned meta-analysis considered all studies with patients above >500 mg/dL), then the association will be less clear. In our opinion, this is new indirect proof that severity might be closely linked to serum TG levels.

A significant statistical association between HTG-AP and ICU admissions was observed in our study, as well as a meaningful association with longer median LoS in ICU (7.0 days in HTG-AP vs. 3.0 days in OAP). Although all studies reviewed [24,30] showed a link between HTG etiology and ICU admission, we found only limited data concerning the LoS in ICU. One retrospective study [29] found a median LoS-ICU of 4.0 days (IQR = 12.0) shorter than in our study. Corroborated with the previous findings regarding severity, we can consider that these two results are other indirect proofs that HTG-AP might be closely related to a more severe course of the disease, probably TG level-dependent [31].

We observed a statistically significant association between younger age and HTG-AP, the median age in our study being 44.5 (IQR = 9). This finding is consistent with the published data [24,27,32,33,34]. However, in a systematic review [16], age was not correlated with HTP-AP. Perhaps the association with younger age is due to a high-fat diet, as older age is often associated with a restriction on fat intake, as cardiovascular disorders require. Another possible cause might reside in lower healthcare adherence in the younger population.

Regarding the general median LoS, we found that HTG-AP is related to statistically significant longer stays (8.3 days vs. 7.0 days in the OAP group). This finding is contrary to that of a retrospective study from Turkey [35] with a smaller population but similar to that of the APPRENTICE multi-centric prospective study [24].

In our cohort, HTG-AP seems to be linked to the development of APFC and is less likely to leave the pancreas with no macroscopical alterations. However, the main limitation of our study is the limited available data on the morphology of the pancreas (69.7% availability). An association between a higher rate of local complications and HTG-AP is described in the literature. Nevertheless, there is a heterogeneity in the published results: some findings [36] indicate an association with necrosis, while others [13,29] indicate an association with both necrosis and fluid complications and yet others [25], like our study, find a tie between HTG-AP and APFC.

One limitation of our study indeed is that we did not split the HTG-AP group into smaller groups, as Mosztbacher et al. did. In our opinion, any association between APFC and HTG-AP should be studied further, as there is a high probability of evolving into pseudocysts, a complication that has a high risk of evolving further to a pseudoaneurysm, a complication presenting a vital risk and hard to diagnose and treat [37].

Unsurprisingly, in the HTG-AP group, there was a statistically significant predominance of cases treated in gastroenterological services. We could not find any previous data in the literature on this particular topic, but we do not completely exclude that there might be some sources in this regard.

Mosztbacher et al. [13] found, in a prospective international cohort, a correlation between male gender and HTG-AP, a correlation that we did not observe in our population, although proportionally there were more males in the HTG-AP group.

Even if we found a slightly higher mortality rate in the HTG-AP group (5.2% vs. 4.4%), we did not find any meaningful disparities, similar to another retrospective study from 2022 [31] and a prospective one [24]. Still, our findings are opposed to those of a 2017 meta-analysis [38] that showed a link between a higher probability of mortality in the presence of HTG-AP. This conflict of results, we believe, resides in the design of the study more than in the real-life setting of our population.

Similar to the baseline characteristics of the population we studied, in the HTG-AP group we found a predominance of urban cases (70.7%), and no important disparities when compared with the OAP group. In a New Zealand study [39], the ratio of urban/rural patients indicated a unitary number but a study from Albania [40] revealed a similar pattern of rurality as ours. We believe this result is linked most likely to the level of development of healthcare facilities and to cultural factors rather than to an environmental risk for AP in general.

We found a slightly lower median DCH for HTG-AP (167.5 EUR vs. 186.9 EUR) but we did not find a statistically significant difference between the two groups. Higher costs associated with HTG-AP were found by both USA [41] and Chinese [42] studies. Differences might reside in the fact that the USA has a different organization of the healthcare system, as the main payer is the patient, while the Chinese study compared HTG-AP only with biliary-caused AP, not with all other causes.

Smoking is considered an independent risk factor in alcoholic AP [43] but we could not find any significant differences between the two studied groups (probably due to missing data on smoking status). However, other retrospective studies [36,44,45] also did not find any association between HTG-AP and tobacco smoking. We consider that one of the limitations of our study is a large number of missing data regarding tobacco smoking and as such we encourage future studies on this particular subject.

Our study has several limitations due to its retrospective design, large percentage of missing data in EHRs on several aims (Table 1) and the arbitrary threshold for serum TG used. As strengths of this investigation, we consider the following: the large population (*n* = 1855), the time efficiency, as the last discharge in this study was roughly one year prior to drafting this article, and the real-world clinical practice origin of the data.

## 5. Conclusions

In this study, we found that the typical case study of HTG-AP is as follows: a middle-aged patient, with previous episodes of AP, that will be cared for in a gastroenterological ward with a longer ICU stay and hospitalization, who is less likely to have a mild course of disease and has a higher probability of developing APFC. We advocate for the standardization of the cut-off of serum TG in the definition of HTG-AP.

## Figures and Tables

**Figure 1 diagnostics-14-01105-f001:**
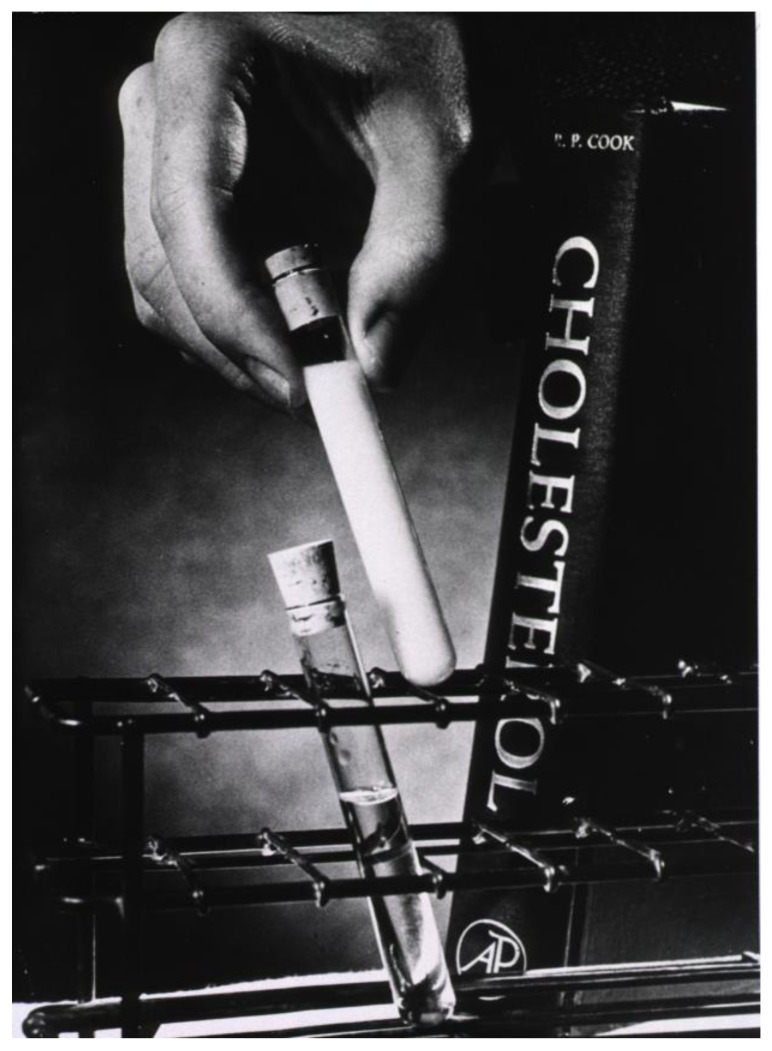
The milky serum of a patient with dyslipidaemia as photographed by J. Hecht in the 1960s (courtesy of the National Library of Medicine).

**Table 1 diagnostics-14-01105-t001:** Population characteristics.

AP cases	1855
**Recurrence**	
Acute pancreatitis (first known attack)	1536 (82.8%)
Recurrent AP	319 (17.2%)
**Age (years)**	
Median	57 (IQR = 26.0)
Mean	56.9 (SD = 17.1)
**Days of hospitalization**	
Median	7.0 (IQR = 6.0)
Mean	8.8 (SD = 7.8)
**Length of stay ICU (days)**	
Median	3.0 (SD = 5.4)
Mean	5.2 (IQR = 5.0)
**Daily cost of hospitalization (EUR)**	
Median	184.9 (IQR = 86.8)
Mean	432.3 (SD = 3439.7)
**Aetiology**	
Hypertriglyceridemia	58 (3.1%)
Idiopathic	283 (15.2%)
All other known causes, out of which:	1514 (81.7%)
Biliary	732 (39.5%)
Alcoholic	628 (33.9%)
Diabetes mellitus	55(3.0%)
Pharmacological	30 (1.6%)
Trauma	19 (1.0%)
Other known causes	50 (2.7%)
**Sex**	
Male	1098 (59.2%)
Female	757 (40.8%)
**Severity**	
Mild	954 (51.4%)
Moderately severe	677 (36.5%)
Severe	224 (12.1%)
**Morphology**	
Interstitial	715 (38.5%)
Normal pancreas	274 (14.8%)
APFC	136 (7.3%)
ANC	87 (4.7%)
Pseudocyst	76 (4.1%)
WON	5 (0.3%)
No data	562 (30.3%)
**Outcome**	
Healed/ameliorated	1540 (83.0%)
Discharge-at-will	116 (6.3%)
Deceased	108 (5.8%)
Transferred	79 (4.3%)
Stationary	12 (0.6%)
**ICU**	
No	1676 (90.4%)
Yes	179 (9.6%)
**Tobacco smoking**	
Active	324 (17.5%)
Former (>4 weeks)	90 (4.9%)
Never	35 (1.9%)
No data	1406 (75.8%)
**Ward of origin**	
Gastroenterology	941 (50.7%)
Surgery	914 (49.3%)
**Place of origin**	
Urban	1332 (71.8%)
Rural	507 (27.3%)
No data	16 (0.9%)

**Table 2 diagnostics-14-01105-t002:** Clinical and demographical characteristics of HTG-AP vs. OAP.

	HTG-AP (*n* = 58)	OAP (*n* = 1514)	*p*-Value
Severity			
Mild	19 (32.8%)	**796 (52.6%)**	*p* < 0.01
Moderately severe	28 (48.3%)	551 (36.4%)	
Severe	11 (19.0%)	167 (11.0%)	
ICU			
No	48 (82.8%)	1389 (91.7%)	*p* = 0.02
Yes	**10 (17.2%)**	125 (8.3%)	
Ward of care			
Gastroenterology	**42 (72.4%)**	744 (49.1%)	*p* < 0.01
Surgery	16 (27.6%)	770 (50.9%)	
Gender			
Male	37 (63.8%)	917 (60.6%)	*p* = 0.62
Female	21 (36.2%)	597 (39.4%)	
Outcome			
Healed/ameliorated	48 (82.8%)	1270 (83.9%)	*p* = 0.63
Stationary	1 (1.7%)	10 (0.7%)	
Transfer	1 (1.7%)	72 (4.8%)	
Discharge at will	5 (8.6%)	95 (6.3%)	
Deceased	3 (5.2%)	67 (4.4%)	
Recurrence			
First attack	40 (69.0%)	1249 (82.5%)	*p* < 0.01
Recurrence	**18 (31.0%)**	265 (17.5%)	
Morphology/local complications			
Interstitial	26 (44.8%)	597 (39.4%)	*p* = 0.02
APFC	**9 (15.5%)**	110 (7.3%)	
Pseudocyst	4 (6.9%)	56 (3.7%)	
ANC	3 (5.2%)	58 (3.8%)	
WON	0 (0.0%)	4 (0.3%)	
Normal pancreas	1 (1.7%)	**241 (15.9%)**	
No data	15 (25.9%)	448 (29.6%)	
Smoking tobacco			
Active	9 (15.5%)	283 (18.7%)	*p* = 0.09
Former	7 (12.1%)	74 (4.9%)	
Never	1 (1.7%)	25 (1.7%)	
No data	41 (70.7%)	1132 (74.7%)	
Rurality			
Urban	41 (70.7%)	1073 (70.9%)	*p* = 0.74
Rural	17 (29.3%)	426 (28.1%)	
No data	0 (0.0%)	15 (1.0%)	
Age (years)			
Mean	44.3 (SD = 7.8)	57.2 (SD = 17.0)	*p* < 0.01
Median	44.5 (IQR = 9)	57.0 (IQR = 25)	
Length of stay (days)			
Mean	11.2 (SD = 8.3)	8.7 (SD = 7.2)	*p* = 0.01
Median	8.3 (IQR = 9.3)	7.0 (IQR = 5.0)	
Length of stay ICU (days)			
Mean	12.5 (SD = 11.4)	4.6 (SD = 4.1)	*p* < 0.01
Median	7.0 (IQR = 22.5)	3.0 (IQR = 4.0)	
Daily Cost of Hospitalization (RON)			
Mean	210.7 (SD = 121.76)	439.1 (SD = 3556.7)	*p* = 0.59
Median	167.6 (IQR = 101.4)	186.9 (IQR = 84.1)	

## Data Availability

Additional data available upon reasonable request from the corresponding author.

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
