# Peer review of "Hypertriglyceridemia-Induced Acute Pancreatitis—The Milky Way Constellation—The Seven-Year Experience of a Large Tertiary Centre"

_diagnostics, 2024, doi:10.3390/diagnostics14111105_

Round 1
Reviewer 1 Report
Comments and Suggestions for Authors
See attached file
Line 17 known cause of pancreatitis
Line 17 The research was a retrospective unicentric cohort study
Line 20 The HTG-AP patients were younger
Line 21 than the OAP patients
Line 22 that the HTG-AP patients were more likely
Line 24 Hypertriglyceridemia-induced APs have a more severe course
Line 52 P cases are gradually
line 53 probably dose dependent
what dose dependency are you talking about? There is no drug involved here. May be you mean triglyceride level instead of dose?
line 56 of 26% in the US adult population
line 61-62 ;some sources consider severe any level >1000 mg/dL [10, 15],
while others consider severe any level >500 mg/dL [16, 17].
line 70 in the number of recurrent episodes of pancreatitis when HTG-AP
line 94 image criteria).
line 102 with 1,099 beds
line 124 The cost of hospitalization data is reported in Romanian leu (RON).
I suggest you convert it to dollars or euros
Figures 2 and 3 should be eliminated, They are confusing and do not add anything to the main subject. I do not understand what they are supposed to show.
Figure 4 is also sort of useless.
Barr representation is inadequate to show what the authors want to show, The tables in the paper are quite clear, but the figures are not.
All th Barr charts are confusing because they are comparing different number of patients. HTP and OAP are not comparable in barr charts. it only adds confusion to a manuscript that otherwise is quite clear with the tables.
Line 277 probably dose-dependent
What dose dependency are you talking about?
General suggestion: remove all the barr charts.
Comments on the Quality of English LanguageEnglish corrections are listed in the attached file
Reviewer 2 Report
Comments and Suggestions for Authors
Dear Redactors,
Thank you very much for the possibility to revise the article “Hypertriglyceridemia-Induced Acute Pancreatitis – the Milky Way Constellation – the Seven-Year Experience of a Large Tertiary Centre”.
The article is very interesting and well-written.
I have just a few remarks.
In the introduction section please describe acute pancreatitis epidemiology in more details.
In the methods, add the information about blood testing (was it frozen, what were the methods used).
In the discussion I get the feeling that smoking and its impact on described issue wasn’t discussed enough. Please, make it more clear.
Thanks
